# Influence of the Severity of Osteogenesis Imperfecta on Cranial Measurements [note 1]

**DOI:** 10.3390/children10061029

**Published:** 2023-06-08

**Authors:** Manuel Joaquín De Nova-García, Rafael G. Sola, Laura Burgueño-Torres

**Affiliations:** 1Dental Clinical Specialties Department, Faculty of Dentistry, Complutense University of Madrid, 28040 Madrid, Spain; denova@ucm.es; 2UAM Chair “Innovation in Neurosurgery”, Department of Surgery, Autonomous University of Madrid, 28049 Madrid, Spain

**Keywords:** craniocervical junction, Osteogenesis Imperfecta, cranial base, platybasia, basilar invagination

## Abstract

Osteogenesis Imperfecta (OI) is a disease that causes bone fragility and deformities, affecting both the cranial base and the craniocervical junction, and may lead to other neurological disorders. A retrospective cross-sectional study was carried out based on cephalometric analysis of the cranial base in a sample of patients with OI, in lateral skull radiographs and magnetic resonance imaging (MRI), comparing them with a sample of age-matched controls. When the different variables of the craniocervical junction were analyzed, significance was found in comparisons with the different age groups. All measurements of the variables studied stabilized as growth progressed. For most of the variables, the severity of the disease influences the measurements of the skull base, with statistically significant differences. Both age and severity of the disease are factors that directly influence the anatomy of the craniocervical junction in these patients and may serve as indicators in the early detection and prevention of other derived alterations.

## 1. Introduction

The craniocervical junction (CCJ), also known as the occipitocervical or craniovertebral junction, is a complex transition between the skull and the cervical spine, and the brain and spinal cord, respectively. It is constituted by the occipital bone and the first two cervical vertebrae. Thus, bone abnormalities in CCJ may involve not only bone structures but also the central nervous system. Nerve structures related to the CCJ include the cerebellum, the fourth ventricle, the caudal portion of the brainstem, four lower cranial nerves, and the adjacent subarachnoid space [1,2].

The first cervical vertebra, Atlas, is ring-shaped and articulates superiorly with the occipital through two lateral bodies. The central hollow serves to give way to the spinal cord and also articulates with the axis tooth. Axis is the second cervical vertebra, whose main characteristic is that it presents in its superior face a voluminous vertical eminence, the axis tooth, also called odontoid process, which articulates in the anterior arch of the atlas. This hinge is responsible for the proper movement of the head on the neck. The unique combination of mobility and stability of the CCJ is provided by strong ligamentous structures that connect the atlas to the axis and the occipital bone.

Osteogenesis Imperfecta (OI) is a congenital hereditary disorder characterized by the presence of osteopenia and increased bone fragility [3,4]. The literature reports alterations of the cranial base in affected patients, producing a flattening of the anterior cranial base that results in the approach of the odontoid process to the foramen magnum. Several neurological symptoms associated with brainstem compression, which may even lead to death, have been described [5]. Muscle weakness is part of the pathophysiology of OI [6], contributing to increased skeletal deformity during growth.

The origin of the CCJ anomalies is still unknown, although it has been associated with repeated microfractures in the region of the foramen magnum as well as softening of the bone, resulting in invagination of the occipital condyles and decreasing height of the skull base, defined as basilar impression or basilar invagination [7]. Thus, the odontoid process of the axis restricts the space in the posterior fossa and may cause compression of the brain stem and spinal cord at the foramen magnum, with consequent neurological alterations [5,8,9,10].

CCJ anomalies are more common in the most severe forms of OI [7,8,11]. While basilar invagination and basilar impression develop throughout life due to a softening of the bony structures of the skull base, platybasia is the most frequent anomaly in patients with OI, appearing either in isolation or in association with the two previous ones [5,8,9,10]. Although these alterations do not always present symptomatology [12], adolescence (11–15 years) seems to be the most common age of presentation of these alterations since this is when basilar impression progresses most [9]. CCJ alterations have been studied as a cause of death in cases of patients with OI, finding in the case of type III and V OI a high percentage of basilar invagination and respiratory deaths [3,13]. In order to avoid or eliminate the associated neurological symptoms, the literature includes some surgical techniques such as posterior fossa decompression with or without instrumentation, transoral or endonasal decompression with posterior occipitocervical fission, or gravity traction of the halo with posterior instrumentation [5,14,15].

The neurological symptoms that a patient may present as a result of a skull base anomaly are very diverse: nystagmus, dysphagia, hearing loss, vertigo, sleep apnea, convulsions, headaches caused by coughing/laughter, loss of sensation in the legs or arms, and alterations in consciousness, among others [4,8,9,16,17,18]. These symptoms are frequently progressive, leading to neurological deterioration, respiratory distress, or even death [5,8,9]. Thus, it may be possible to anticipate which children require closer neurological monitoring by studying the anatomy of the cranial base from their lateral skull radiographs and brain MRIs and performing craniometry from anatomical landmarks.

Therefore, the objective is to study the craniocervical junction in patients with OI, with different degrees of severity, analyzing changes in the CCJ at different ages.

## 2. Materials and Methods

### 2.1. Study Design

After obtaining approval from the Clinical Research Ethics Committee of San Carlos Clinic Hospital (C.P.-C.I. 13/033-E; approved on 22 February 2013) and the University Hospital of Getafe (A07-15; approved on 3 June 2015), a retrospective cross-sectional study was carried out based on cephalometric analysis of the skull base in patients with OI and comparisons were made with a sample of age-matched controls.

### 2.2. Study Sample and Controls

To be included in the study sample, all patients had to be minors, present a previous MRI or teleradiography, and have parental/guardian consent. We excluded patients whose imaging tests had not followed a standardized protocol or were of insufficient quality for analysis.

In the case of the control sample, two age-matched healthy controls were taken for each patient in the study sample. MRI scans were selected from the database of the Radiology Department of the University Hospital of Getafe (Madrid), taking into account that the patients did not have diseases related to alterations in the craniocervical junction or congenital anomalies of the skull base. The lateral skull radiographs were selected from the database of an Oral and Facial Diagnostic Center, taking into account that the patients had skeletal class I, since a relationship has been found between skull base anomalies and vertebral anomalies in patients with class III [5].

The selection of these patients was carried out by an independent researcher blind to the objectives of this research, who codified the name of each patient according to the Spanish Data Protection Law.

### 2.3. Data Collection

MRI scans were performed without contrast following a standardized protocol and the patients were placed in supine decubitus with the head in a neutral position. All the working images were reviewed and the slice that best visualized the tip of the odontoid process was selected, with the distances and angles in this slice being calculated. The lateral skull radiographs were obtained using a standardized protocol, placing the patient’s head in the positioner with the Frankfurt plane parallel to the floor.

All measurements were performed using the RadiAnt DICOM Viewer software version 2.0.12. Two examiners analyzed the images. The first examiner performed a cephalometric analysis of all the images (42 MRI and 57 lateral skull radiographs) following a random order and then performed the same analysis after 15 days, thus determining intra-examiner efficiency. The second examiner evaluated 10 MRIs and 10 lateral skull radiographs randomly selected from the entire sample (OI and controls), to calculate interobserver agreement.

For all imaging tests, eight linear (McRae, Chamberlain, modified McGregor, Kovero, Wackenheim, Ranawat, modified Ranawat, and Redlund-Johnell) and five angular (Arponen, craniovertebral, clivus-canal, basal, and Boogard) measurements were plotted to determine the presence of craniocervical junction alterations.

All distances and angles analyzed are shown in Figure 1.

### 2.4. Statistical Analysis

Statistical analysis of the data was performed using the SPSS program version 22 for Windows, establishing a significance level of *p* < 0.05.

-Descriptive statistics of quantitative and qualitative variables.-Student’s *t* test for the comparison of two means; Mann–Whitney U test for non-normal distributions; Shapiro–Wilk test to determine normality.-ANOVA for the comparison of multiple means. In the case of non-homogeneous variances, robust tests of equality of means were used: the Welch and Brown–Forsythe tests. When the data did not come from a normal distribution, we used the Kruskal–Wallis test to compare multiple means.-Pearson correlation test to study the correlation with age, applying regression lines and analysis of covariance (ANCOVA), to the variables where correlation with age was observed.

## 3. Results

Considering all inclusion and exclusion criteria, the final study sample consisted of 33 images, 14 MRIs, and 19 lateral skull radiographs of 28 patients with OI.

When analyzing intra- and interobserver agreements, an almost perfect intraclass correlation coefficient (0.81–1.00) was obtained in the intraobserver coefficient for almost all variables, with the exception of the McRae line, Ranawat line, and Boogard angle, located in the lateral skull radiographs, in which cases the agreement was substantial (0.79, 0.79, and 0.8 respectively); these difficulties were solved by performing the analysis using an MRI. The interobserver agreement was almost perfect in all the variables, obtaining indices between 0.81 and 1.00.

When analyzing the CCJ measurements, both the control and study samples were grouped by age, since some of these measurements change with growth. This is the case of the Ranawat variable analyzed in lateral skull radiographs in the control sample (*p* = 0.00), in which we observed differences between the 6–8 age group and the rest, with stabilization after 9 years of age. The same was the case for the modified Ranawat variable studied in lateral skull radiographs in the control samples (*p* = 0.00), in which the differences were in the 6–8 and 9–11 age groups with respect to the rest, with stabilization from the age of 12. The Redlund-Johnell measurement also showed differences in the lateral skull radiographs (*p* = 0.00) between the 6–8, 12–14, and 15–18 age groups and between the 9–11 and 12–14 age groups within the control samples. This measurement increases with age and stabilizes between 12 and 18 years of age. In the case of the angle of the base of the skull, a negative correlation, decreasing with age, was found.

### 3.1. CCJ Measurements on MRI

When evaluating the differences between the OI study group and the controls, the first thing we observed is that in most of the variables, the severity of the disease significantly influences the skull base measurements.

When studying the variables analyzed using the MRI (Table 1), differences were observed in the Wackenheim measurement between the control groups and patients with moderate OI (*p* = 0.023) and between patients with mild and moderate OI (0.041). In the Ranawat and modified Ranawat variables, the differences were between the severe and moderate OI groups and the control group. In the Redlund-Johnell method, differences between groups were found between the control group of healthy patients and patients with severe OI (*p* = 0.001). In the skull base angle (N-S-Ba), differences were found between the control and the other groups and between the severe OI and the other groups. In the clivus-canal angle, differences were observed between the control group and patients with mild OI (*p* < 0.0001). In the Boogard angle, the differences between groups were between the control group of healthy patients and patients with severe OI (*p* = 0.002).

### 3.2. CCJ Measurements on Lateral Skull Radiographs

In the variables studied in the lateral skull radiographs (Table 2), statistically significant differences were observed in the Ranawat variable between the control and III and control and IV groups, and between the mild OI and severe OI groups. In the modified Ranawat measure, statistically significant differences were found between group III and the control group. In the Redlund-Johnell method, the greatest differences were observed between the control group and severe OI, although these were not statistically significant. The angle of the skull base shows significant differences between the control group and the three types of OI. The craniovertebral angle showed differences between the control and mild OI groups and the mild and severe OI groups. The clivus-canal angle showed differences between the moderate OI and the control groups.

## 4. Discussion

The diagnostic tools of choice for the confirmation of pathology at the CCJ level are computed tomography (CT) and magnetic resonance imaging (MRI), with the latter procedure considered to offer the highest diagnostic sensitivity and specificity [10,19,20]. Even so, lateral skull radiographs are recommended for the purpose of performing a simple, economical, and low-radiation initial evaluation in patients at risk, this being the case for patients with OI [20,21].

In addition to the variables studied in the present study, others can be found in the literature, including the omega angle, Welcher’s basal angle, and Fischgold lines, among others [10,16,22,23]; however, they were not included in this study because their relationship in OI patients has not been documented.

The reliability of measurements in conventional radiographs is questioned by some authors [24] since it is subject to the localization of cephalometric points by the examiner. This is why some authors have analyzed this variable in more depth. In 2008, Arponen et al. [21] analyzed the differences between two examiners when locating certain cephalometric points in lateral skull radiographs, concluding that the basion and opisthion points are the most difficult to locate (affecting the McRae and Chamberlain lines and the anterior cranial base angle), although these difficulties can be overcome using the McGregor line, the measurement of which is more reproducible. Despite these differences, they agreed that they are of low clinical significance. In 2011, Kwong et al. [22] evaluated 200 CT scans and concluded that the Wackenheim line is the least reliable of the measurements. The present study analyzed the reproducibility of the measurements involved, not of the isolated cephalometric points, and found agreement of 0.81–1.00 between both examiners for almost all variables, with somewhat lower coefficients in the measurements in which there is more overlapping of structures in the radiographs (basion, opisthion, and axis sclerotic ring). However, despite this, the coefficients obtained when analyzing the differences between both examiners were acceptable.

Lateral skull radiography is performed in patients with susceptibility to skull base anomalies as an initial evaluation of pathology, but it has the main disadvantage of overlapping structures, which makes it difficult to locate certain points, mainly those that delimit the foramen magnum (Basion and Opisthion) and the lowest point of the occipital scale. It is also complicated to locate the center of the sclerotic ring of the axis, necessary for the Ranawat line, which is solved with the modified Ranawat line, which changes this point to the midpoint of the inferior border of the odontoid process of the axis. Discrepancies are corrected when taking measurements using MRI, since the measurements are more reproducible due to the avoidance of overlapping structures. However, in MRI, due to the positioning of the patient’s head, there are points that may not be located in the selected slice. In the present study, authors selected the section in which the most superior point of the odontoid process of the axis, the reference for most of the measurements, was best seen. Difficulties were encountered when locating the most anterior point of the frontonasal suture (Nasion) in the same slice. Therefore, it is important to have all the MRI slices, which serve as an orientation to locate the reference point.

Few works study CCJ anomalies in patients with OI and, moreover, there is no uniformity in the diagnostic criteria. The references and measurements used by the different authors are variable [7,16,25]. Kovero et al. [21] proposed new diagnostic criteria based on the mean and standard deviation of their healthy controls and took three standard deviations (SD) above the mean as the limit of pathology. Cheung et al. [26] found skull base anomalies in all types of OI, which corresponds with the data in the present study. During facial growth, changes in the palate, its position in relation to the craniocervical junction, and as a consequence, the Chamberlain and McGregor line measurements (which use the hard palate as a cephalometric point), increase with age. The Kovero line changes less with age because it uses the anterior cranial base as a reference structure, which stabilizes early during midline growth. The anatomy and variations that occur in the base of the skull as well as in the CCJ are also associated with the severity of the malocclusion present in patients with OI [27,28] such that in cases of type III and IV OI, the involvement, as well as the progression of malocclusion during growth, will be greater [29].

The present study did not find a linear relationship between the age of the patients and the severity of OI; however, the type of OI diagnosed was a determining variable in the changes observed in the cranial measurements in relation to age. In accordance with these results, in 2010, Arponen et al. [30] demonstrated that age is relevant in growing subjects when assessing skull base anatomy. They provided specific age-appropriate normal reference values from age 4 for the anterior skull base angle, as well as for the Chamberlain, McGregor, and Kovero measurements. A range of ±2.5 standard deviations is considered an appropriate cutoff point for this purpose. In addition, they note that with age, the odontoid process approaches the base of the skull in the Chamberlain, McGregor, and McRae measurements, but not so with the Kovero line. This line is not age-dependent, thus a marked change in this measurement may be a good indicator of abnormal individual development. A few years later, in 2015 [31] they determined in a study of 39 patients with OI who were aged between 0 and 25, that 25% of the patients with mild OI exhibited a skull base anomaly. Of those with moderate OI, 70% had an anomaly, and of those with severe OI, 78% had an anomaly. In this study, 8 patients had type I OI (slight), and 50% of them had CCJ abnormalities; of the 11 patients with type III OI (moderate), 10 had abnormalities; and of those with type IV OI (severe), 30% had alterations (9 patients in total).

The reported prevalence of craniocervical abnormalities is higher in adults with OI than in children [20,26], indicating progression over time. Furthermore, in most of the variables analyzed in the present study, the severity of the disease influences the measurements of the CCJ, showing a tendency for the odontoid process to approach the base of the skull to a greater extent as the severity of OI increases (type III and IV OI). By comparing the study and control samples, it was observed that the CCJ measurements were altered and statistically significant differences were found in the McRae, Chamberlain, Wackenheim, Ranawat, modified Ranawat, Redlund-Johnell, craniovertebral angle, clivus-canal, basal, and Arponen measurements. The greatest differences were found between the control group and the most severe forms of the disease (types III and IV). In 2021, Ludwig et al. [3] report 19% of cases of CCJ alterations in their sample of 37 patients with type V OI. However, this cannot be compared with the present study because they did not present in their sample patients with type V OI.

Studies relating other clinical characteristics of the disease and its treatment with skull base anomalies report possible relationship with height (lower z-score), irrespective of the treatment received [3,32]. Additionally, a higher risk of basilar impression or invagination in severe forms of OI, which would not be avoided with bisphosphonate treatment, although its earlier onset could delay the development of the pathology [31]. Regarding the natural course of these anomalies, beyond the empirical appreciation expressed by clinicians, Arponen et al. [11] have confirmed that there is no evidence of their progressive nature.

Sillence [12] recommends radiographic evaluation every 2–3 years from 5 years of age in severe forms of OI and suggests radiographic analysis of skull base dimensions in all patients before school age. In severe forms of OI, radiography or MRI is often performed in the early ages in order to diagnose the type of OI, and in these cases, MRI can be used to evaluate the craniocervical junction. If no abnormality is observed on the image(s) obtained in the early ages, further imaging would not be indicated in the case of asymptomatic patients, thus reducing radiation for the patient. In case of abnormal findings, an individual follow-up and treatment plan should be justified [11].

## 5. Conclusions

After the study, the authors concluded that the greatest pathology of the skull is found in the most severe forms of the disease, that is, OI types III and IV. When studying the base of the skull in growing patients, it is observed that in the case of the Ranawat line, the modified Ranawat line, and the Redlund-Johnell method, there is a positive correlation with age; on the other hand, the angle of the anterior cranial base has a negative correlation with age, with statistically significant results.

In the samples studied, a tendency of the odontoid process to approach the base of the skull with the severity of the disease can be observed in patients with OI (type III is more severe than type IV, and this in turn is more severe than type I). In the same way, with the angular measurements, it is determined that the anterior cranial base flattens with the severity of the disease (type III > IV > I), with statistically significant differences in the McRae, Chamberlain, Wackenheim, Ranawat, modified Ranawat, Redlund-Johnell, craniovertebral angle, clivus-canal, basal, and Arponen measurements.

Lateral skull radiography and the variables studied provide a good tool to anticipate CCJ problems in patients with OI. If abnormal findings are suspected, or pathology is present, an MRI would be warranted.

## Figures and Tables

**Figure 1 children-10-01029-f001:**
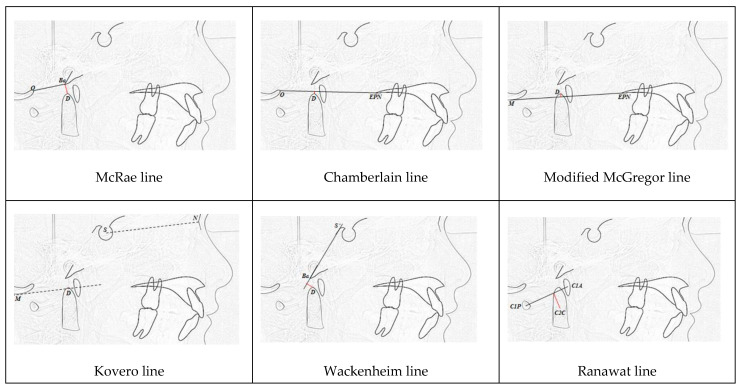
Lines and angles analyzed in craniometry.

**Table 1 children-10-01029-t001:** Comparative analysis of craniocervical junction measurements on MRI between the different types of OI and controls. * *p* < 0.05.

CCJ Measurements	Controls (*n* = 28)	OI (*n* = 14)			*p*
		Slight (*n* = 4)	Moderate (*n* = 4)	Severe (*n* = 6)	
McRae	−4.87 ± 1.24	−4.58 ± 0.77	−3.2 ± 1	−1.69 ± 4.31	0.016 *
Chamberlain	−2.26 ± 3.23	−0.6 ± 2.05	1.95 ± 2.48	4.96 ± 7.35	0.033
Modified McGregor	−1.16 ± 3.33	0.34 ± 2.55	1.76 ± 3.05	10.06 ± 12.81	0.068
Kovero	−5.57 ± 4.23	−6.48 ± 2.07	−5.13 ± 5.03	5.24 ± 17.19	0.193
Wackenheim	−2.2 ± 1.52	−2.04 ± 0.56	0.38 ± 0.93	0.48 ± 4.49	0.008 *
Ranawat	14.99 ± 2.04	13.8 ± 1.67	11.91 ± 1.08	12.2 ± 1.88	0.003 *
Modified Ranawat	26.81 ± 2.82	25.08 ± 1.37	23.45 ± 1.36	23.34 ± 1.83	0.007 *
Redlund-Johnell	34.37 ± 4.32	31.19 ± 3.74	29.54 ± 4.45	19.07 ± 13.17	0.001 *
Arponen	9.76 ± 4.91	9.44 ± 3.59	6.4 ± 5.62	17.03 ± 13.65	0.072
Cranio-vertebral	90.66 ± 7.76	86.43 ± 1.36	91.95 ± 8.41	89.19 ± 13.15	0.768
Clivus-canal	153.62 ± 7.47	141.23 ± 2.5	150.41 ± 5.3	135.86 ± 20.47	0.001 *
Basal	129.71 ± 6.45	139.1 ± 4.17	138.06 ± 6.86	151.25 ± 8.53	0.000 *
Boogard	118.22 ± 6.24	126.05 ± 6.48	123.51 ± 6.34	140.63 ± 18.9	0.001 *

**Table 2 children-10-01029-t002:** Comparative analysis of craniocervical junction measurements in lateral skull radiographs between different types of OI and controls. * *p* < 0.05.

CCJ Measurements	Controls (*n* = 38)	OI (*n* = 14)			*p*
		Slight (*n* = 4)	Moderate (*n* = 7)	Severe (*n* = 8)	
McRae	−4.54 ± 2.56	−5.04 ± 2.58	−3.99 ± 2.04	−1.93 ± 4.45	0.292
Chamberlain	−1.72 ± 2.98	−2.85 ± 4.24	−0.6 ± 3.26	−0.12 ± 5.1	0.639
Modified McGregor	0.13 ± 3	−0.81 ± 4.65	1.54 ± 2.89	2.12 ± 4.59	0.442
Kovero	−4.52 ± 3.47	−7.1 ± 6.91	−5.61 ± 5.05	−5.57 ± 7.35	0.909
Wackenheim	−3.27 ± 2.32	−4.25 ± 2.29	−1.84 ± 1.24	−2.84 ± 3.33	0.374
Ranawat	14.33 ± 1.95	12.49 ± 1.96	12.55 ± 1.04	10.08 ± 1.99	0.000 *
Modified Ranawat	26.35 ± 3.08	23.41 ± 3	24.94 ± 1.57	22.59 ± 2.36	0.000 *
Redlund-Johnell	33.12 ± 5	29.68 ± 6.55	30.27 ± 4.6	28.79 ± 5.66	0.000 *
Arponen	8.47 ± 4.96	13 ± 12.52	10.07 ± 6.76	12.88 ± 8.39	0.534
Cranio-vertebral	92.94 ± 7.45	90.43 ± 2.51	93.55 ± 11.66	102.65 ± 7.98	0.018 *
Clivus-canal	150.26 ± 8.35	140.24 ± 4.19	139.62 ± 13.17	147.45 ± 7.22	0.011 *
Basal	133.06 ± 4.87	143.05 ± 6.38	144.61 ± 4.42	145.43 ± 3.21	0.000 *
Boogard	121.66 ± 6.36	127.14 ± 13.02	128.08 ± 11.8	127.03 ± 10.17	0.109

## Data Availability

The data presented in this study are available on request from the corresponding author upon reasonable request.

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
