# Peer review of "Influence of the Severity of Osteogenesis Imperfecta on Cranial Measurementsâ€"

_children, 2023, doi:10.3390/children10061029_

Round 1
Reviewer 1 Report
1. Patients with OI treated with bisphosphonate are a common treatment plan. However, bisphosphonates can cause severe adverse effects like bone, joint, and muscle pain. In that case, how it affects this study?
2. Craniocervical abnormalities in osteogenesis imperfect what is the Genetic and molecular correlation?
3. Craniocervical Junction MRI scans can determine if craniocervical instability is present. This is a condition that can cause constant headaches and a heavy head feeling. Is there any craniocervical instability reported in your samples?
1. Patients with OI treated with bisphosphonate are a common treatment plan. However, bisphosphonates can cause severe adverse effects like bone, joint, and muscle pain. In that case, how it affects this study?
2. Craniocervical abnormalities in osteogenesis imperfect what is the Genetic and molecular correlation?
3. Craniocervical Junction MRI scans can determine if craniocervical instability is present. This is a condition that can cause constant headaches and a heavy head feeling. Is there any craniocervical instability reported in your samples?
Author Response
Consulte el archivo adjunto

Reviewer 2 Report
The paper "Influence of the severity of Osteogenesis Imperfecta on the craniocervical junction"describes more the changes in cranial measurements, rather than theie influence on Osteogenesis Imperfecta.
I suggest changing the title.
There is an inconsistency between the aim:
"the objective is to study the craniocervical junction in patients with OI treated with bisphosphonates"
and the conclusions:
"the greatest pathology of the skull is found in the most severe forms of the disease, that is, OI types III and IV"
So, where is the connection between bisphosphonates and OI?
The clinical relevance must be added, in the discussion as well as in the conclusion.
Some real MRI images must be shown, as well as the type of cephalogram.
The sample size is confusing: "the final study sample consisted of 33 images, 14 MRIs and 19 lateral skull radiographs of 28 patients with OI".
It would be beneficial to include only subjects that had both: MRI and cephalogram. So maybe only 14 subjects.
A NNT (number needed to treat) must be calculated.
What about the control group?
How was the OI diagnosed?
Subjects must be categorized according to age groups, so maybe 14 subjects would not be sufficient. Please define exactly the term "growing patients" - it is confusing.
I suggest adding more subjects to the study in each age group so that the findings should be relevant to the literature.
Round 2
Reviewer 2 Report
The paper has been improved